# Performance of a Radio-Frequency Two-Photon Atomic Magnetometer in Different Magnetic Induction Measurement Geometries

**DOI:** 10.3390/s24206657

**Published:** 2024-10-16

**Authors:** Lucas Martin Rushton, Laura Mae Ellis, Jake David Zipfel, Patrick Bevington, Witold Chalupczak

**Affiliations:** National Physical Laboratory, Hampton Road, Teddington TW11 0LW, UK; lucas.rushton@npl.co.uk (L.M.R.); laura.ellis@npl.co.uk (L.M.E.); jake.zipfel@npl.co.uk (J.D.Z.); witold.chalupczak@npl.co.uk (W.C.)

**Keywords:** atomic magnetometer, non-destructive testing, magnetic induction tomography

## Abstract

Measurements monitoring the inductive coupling between oscillating radio-frequency magnetic fields and objects of interest create versatile platforms for non-destructive testing. The benefits of ultra-low-frequency measurements, i.e., below 3 kHz, are sometimes outweighed by the fundamental and technical difficulties related to operating pick-up coils or other field sensors in this frequency range. Inductive measurements with the detection based on a two-photon interaction in rf atomic magnetometers address some of these issues as the sensor gains an uplift in its operational frequency. The developments reported here integrate the fundamental and applied aspects of the two-photon process in magnetic induction measurements. In this paper, all the spectral components of the two-photon process are identified, which result from the non-linear interactions between the rf fields and atoms. For the first time, a method for the retrieval of the two-photon phase information, which is critical for inductive measurements, is also demonstrated. Furthermore, a self-compensation configuration is introduced, whereby high-contrast measurements of defects can be obtained due to its insensitivity to the primary field, including using simplified instrumentation for this configuration by producing two rf fields with a single rf coil.

## 1. Introduction

Measurements of the inductive coupling between an oscillating magnetic field and an object of interest represent a well-established technique for the non-destructive testing (NDT) of metalwork. Typically, these measurements are used in the detection of surface features like cracks [1], pitting [2], or measuring variations in coatings [3]. In the generic realisation of inductive measurements (Figure 1), an initial excitation by the so-called primary radio-frequency (rf) field (1) drives the object response, for example the generation of eddy currents in the sample. These in turn generate the secondary rf field (2), which is then monitored by the sensor (3). The variety of names used in reference to this measurement, namely eddy current imaging [4], electromagnetic induction imaging [5], and magnetic induction tomography [6], describe the different origins and properties of the detected signal generation.

In commercial systems, signal detection is typically based on pick-up coils. Although pick-up coils have been demonstrated to have sensitivities at the fT/Hz^1/2^ level, such high-performance sensors are difficult to manufacture, require precision detection electronics, and are optimised for a specific operating frequency. The fundamental sensitivities of pick-up coils are coupled to their volume and operating frequency, limiting the practicality of their miniaturisation at low frequencies.

Magnetic field sensors, such as fluxgates [7] or giant magnetoresistance sensors [8], output a time varying signal constructed from discrete measurements and have been used for ultra-low-frequency (<3 kHz) inductive measurements, but these sensors have limited sensitives at the pT and nT levels, respectively. The rf atomic magnetometer has recently demonstrated 30 aT/Hz^1/2^ sensitivity [9] and is ideally suited for applications in the ultra-low (300 Hz to 3 kHz) to very low (3 kHz to 30 kHz) frequency ranges. Additionally, the rf atomic magnetometer [10,11,12,13,14,15,16,17,18,19,20,21,22] has several unique and desirable features, such as the tunability of the operating frequency [4,10], the ability to obtain the vector measurements of the rf field [16,23,24], sensitivity to rf field polarisation [25,26,27], and high-bandwidth operation in the spin maser mode [28,29,30].

Although effective, many inductive NDT measurements fail to realise the full potential of the inherent non-contact nature of the inductive coupling and penetration of oscillating magnetic fields into the volume of a material, providing potentially deep sub-surface information. The penetration depth is characterised by the skin depth, and, in the low-radio-frequency range (<MHz), for most materials, this is provided by $\delta \propto \sqrt{2/(\omega \sigma \mu })$, where ω, σ, and μ define the rf field angular frequency, electrical conductivity, and magnetic permeability of the medium, respectively. Operating the sensor at low frequencies enables better penetration into the studied structure or through structural barriers. Fundamental limits on sensitivity challenge small inductive detectors like pick-up coils at low frequencies [31]. Similarly, the technical challenges of stabilising the bias magnetic field limit the operation of rf atomic magnetometers at low operating frequencies.

Atomic magnetometers detect rf magnetic fields by optically monitoring the atomic coherences they drive (Figure 2a). The relevant atomic frequency, i.e., the Larmor frequency ($\omega_{0}=\gamma_{\mathrm{cs}}B_{0}$ with $\gamma_{\mathrm{cs}}/\left(2\pi \right)\approx 3.5$ kHz/μT being the gyromagnetic ratio for Cs), is tuned with a static bias field **B**_0_ into resonance with the detected rf field, which has a frequency $\omega_{\mathrm{sp}}$ in the single-photon case [Figure 2a(i)]. The single-photon rf atomic magnetometer is only sensitive to rf magnetic fields $B_{\mathrm{sp}}\left(t\right)$ perpendicular to $B_{0}$ (Figure 2b). To ensure the best sensor performance, the bias field vector needs to be stabilised during the measurement. In unshielded environments, this becomes challenging for bias magnetic fields (or operating frequencies) below 1 μT (3.5 kHz) due to the potentially significant changes in amplitude and direction produced by ambient fields. It is worth reiterating that, in contrast to standard pick-up coils, atomic magnetometers do not suffer from reduced sensitivities at low frequencies. However, the technical challenges of low-frequency measurements in noisy environments triggered searches for alternative solutions.

The benefits of detecting a low-frequency primary rf field $B_{2}\left(t\right)$ with a high sensor operational frequency $\omega_{0}$ can be achieved through the implementation of an auxiliary rf field $B_{1}\left(t\right)$ and detection based on a two-photon process [32,33]. The technique relies on driving atomic coherences by the combination of two rf magnetic fields, where the sum [Figure 2a(ii)] or difference [Figure 2a(iii)] in their frequencies is equal to the sensor’s operating frequency, i.e., $\omega_{0}=\omega_{1}\pm \omega_{2}$ [33]. The interactions between the rf fields and atoms need to meet the momentum conservation $\Delta m_{F}=\pm 1$, which sets a requirement for the polarisations of the rf fields involved in the non-linear process (Figure 2a). From the perspective of the polarised atoms, a circularly polarised rf field (represented by $\sigma^{+}$ in Figure 2a) rotating in the plane perpendicular to the bias field $B_{0}$ provides $\Delta m_{F}=\pm 1$ momentum, whilst a linearly polarised π field, oscillating parallel to $B_{0}$, provides $\Delta m_{F}=0$ (Figure 2c). Thus, when the energy resonance condition is met, the addition of the linearly and circularly polarised fields meets the momentum conservation condition. The two-photon process does not intrinsically improve the magnetic induction or signal generation, but, practically, it is beneficial for the measurement process as it enables the detection of low-frequency fields by the atomic magnetometer without the need to reduce the bias field strength.

While the implementation of the two-photon process was demonstrated in inductive measurements [32], this paper provides systematic studies regarding the technique and the signals generated in magnetically shielded and unshielded environments, and with different coil configurations. We demonstrate several novel aspects with regard to characterising the two-photon transition and utilising it for MIT measurements in the Results section that have not been previously discussed in the literature. (A) The measurements in a magnetically shielded environment enable the identification of all the spectral components resulting from the non-linear interactions between the atoms and rf fields. (B) In inductive measurements, the information about the defect or studied object is contained within both the magnitude and phase of the measured signal. A method for the retrieval of the phase of the two-photon signal is demonstrated. (C) The limitations of the two-photon technique are discussed by comparing the single-photon and two-photon process efficiencies. (D) Studies regarding two-photon detection are extended to inductive measurements and are based on the results recorded in a magnetically unshielded setup with samples mimicking defects within metalwork. An equivalent to the so-called self-compensation arrangement [24] is demonstrated, as well as a simplification of the required instrumentation by producing two rf fields with a single rf coil in the two-photon self-compensation configuration.

This work highlights the strengths and weaknesses of the two-photon method in the applications relevant to inductive measurements. There are geometrical requirements in generating the two-photon coherence and preferential choice regarding the frequencies of the two fields. First, there is an outline of the experimental setup, detailing the recovery of the phase information. Then, the measured signal and its components are described before discussing the efficiency of the two-photon signal relative to the single-photon measurement. The final section details its operation when applied to inductive measurements and explains the relevant configurations to optimise the method’s performance. A general discussion summarising the paper’s findings is provided in the conclusions.

## 2. Experimental Setup

The most important aspects of the experimental setup in this paper involve the generation and detection of the rf fields as the fields required to satisfy a two-photon transition differ from those of the single-photon transition described earlier. For completeness, however, the parts of the experimental setup common to both single- and two-photon transitions are also covered.

It should be noted that experimentally defined and measured frequencies *f* are provided in units of Hz, while ω=2πf is used to denote the angular frequency precessions of the spins and circular rotating rf magnetic fields.

The rf atomic magnetometer sensor consists of three main subsystems: the vapour cell in a magnetically controlled environment, laser, and detection system. The vapour cell, lasers, and detection system together are considered to be the sensor head. Caesium atoms are housed in a 1 cm^3^ cubic glass paraffin-coated vapour cell at ambient temperature ($0.33\times 10^{11}$ atoms). A circularly polarised beam locked to the 6^2^$S_{1/2}$F=3→ 6^2^$P_{3/2}$$F^{\prime }=2$ resonance transition (D2 line, 852 nm) is used to pump the majority of the atoms along a bias magnetic field $B_{0}$ into the $m_{F}=F$ sublevel of the F=4 caesium ground-state level through indirect pumping [34]. As already mentioned, $B_{0}$ defines the energy spitting between the $m_{F}$ sublevels, characterised by the Larmor frequency $\omega_{0}$. A resonant rf field (single- and two-photon conditions described earlier) drives a coherence between $\Delta m_{F}=\pm 1$ sublevels. Oscillations of the atomic coherence amplitude and phase are mapped onto the polarisation of a linearly polarised probe beam propagating orthogonally to $B_{0}$. The probe beam is 2.75 GHz red-detuned from the 6^2^$S_{1/2}$F=3 → 6^2^$P_{3/2}$$F^{\prime }=2$ resonance transition via phase-offset locking to the pump beam. The modulation of the probe beam polarisation is monitored with a simple polarimeter consisting of a half-wave plate, a polarising beam splitter, and a balanced photodetector. Both laser sources are Vescent DBR diodes (D2-100-DBR-852-HP1).

In the shielded setup, the vapour cell sits within three layers of μ metal and an inner layer of ferrite (Twinleaf MS-1LF). The field $B_{0}$ is generated by a set of internal linear and gradient coils and a low-noise current supply (Twinleaf CSB). These coils are also used to produce oscillating magnetic fields $B_{1}\left(t\right)$ and $B_{2}\left(t\right)$. This arrangement is described in Figure 2b,c.

In the unshielded setup, the sensor is operated within a noisy laboratory environment. A feedback control loop (3 × SRS SIM960) is used to stabilise the field measured by a three-axis fluxgate (Bartington Mag690) located close to the vapour cell with three nested orthogonal square Helmholtz coils (1000 mm, 940 mm, and 860 mm side length). The system compensates 50 Hz main electrical noise and drifts in the DC magnetic fields. This unshielded setup is used for inductive measurements (MIT), as shown in Figure 1, with the main components including a set of coils generating the primary rf field (1), the object under investigation (2), and the rf field sensor, i.e., the rf atomic magnetometer (3).

In the standard single-photon rf field configuration, the rf primary field $B_{\mathrm{sp}}\left(t\right)$ is generated by a coil driven by a sinusoidal current with frequency $f_{\mathrm{sp}}$. The coil used in the measurements reported here has 13 turns and an outer-diameter $D_{o}$, inner-diameter $D_{i}$, and length *L* with $D_{o}$:$D_{i}$:L = 5:2:10 mm and is wound on a ferrite core with dimensions of $D_{o}$:L = 2:15 mm, using 0.5 mm diameter copper wire with 0.3 mm thick PTFE coating. Thick-coated wire is used to minimise and accommodate the heating caused by large rf currents. An amplifier is used to drive the coil at currents up to 2 A. For the magnetic induction measurements described in this paper for the single-photon case, the rf field coil axis is centred under the detector (vapour cell) and parallel to $B_{0}$ (Figure 3a). The distance between the coil and the cell is typically 190 mm.

For measurements characterising the two-photon transition, two coils are driven by the signal generator SG1 (Teledyne T3AFG200) (Figure 3b). The generator has two phase-locked outputs: CH1 and CH2 (Phase-locked Mode). Some MIT measurements were performed with two fields produced by a single coil (Figure 3c). In this case, the Wave Combine function of a single channel of SG1 is used such that its output is equal to CH1 + CH2.

The atomic signal measured by the photodiodes can be monitored directly by taking an FFT of a time series with a data acquisition card or a spectrum analyser, or by demodulating the signal into its in-phase and quadrature components using a lock-in amplifier (SRS865). The lock-in has an internally referenced signal generator. The output of this can be used to drive the primary rf coil in the single-frequency (single-photon) case. As mentioned previously, for the two-frequency (two-photon) case, the rf fields are generated by an external signal generator (SG1). The timing of this signal generator is referenced to the Clock Source of a second identical signal generator (SG2). These devices are synchronised using the Multi-Device Synchronisation function, enabling phase locking between the two units. The output CH1 of SG2 generates a signal at $f_{1}\pm f_{2}$, which is used as the external reference to the lock-in amplifier. In this way, the phase of the two-photon coherence can be monitored by the lock-in amplifier.

While in real-life scenarios a fully portable sensor will be moved over the test object, in the laboratory, it is convenient to move the object under the stationary rf coil and sensor. This is achieved with a 2D translation stage with a variable, but typical, step size of ∼0.8 mm. The object studied in this work is a square aluminium plate with cavities drilled in its side to act like a concealed defect (Figure 3).

## 3. Results

### 3.1. Spectral Components by Non-Linear Interactions

The exploration of the two-photon interaction between the rf magnetic fields and the caesium atoms begins with systematic observations of the FFT spectra of the signals generated by atoms driven by two rf fields. This enables observations of all the spectral components of the non-linear interactions. The observations were conducted in a shielded setup to minimise magnetic field noise. However, the same measurements performed in the unshielded setup delivered qualitatively similar results.

Two rf fields are generated by the shield’s internal coils (Figure 2c). For the measurements in this section, the frequency of the auxiliary field $\omega_{1}$ is fixed, while the other, $\omega_{2}$, is scanned over the two-photon resonance.

Figure 4 shows the FFTs of the atomic magneto-optical signal as $f_{2}$ is scanned by 1 kHz around the two-photon resonance frequency. The weak signal around the atomic resonance frequency, $f_{0}=2.48$ kHz, is produced by the atomic projection noise. All the other lines represent the atomic response driven by linear (single-photon) or non-linear (*n* higher-order photon at $f_{0}=f_{1}+nf_{2}$) interactions between the atoms and rf fields. Although most of these interactions are non-resonant, they still produce atomic responses above the noise level defined by the atomic and photonic shot noise. The bright vertical line to the left of the resonance represents linear interactions between atoms and the field with a fixed frequency of $f_{1}=2$ kHz. Non-linear two-photon interactions result in two diagonal spectral components with opposite slopes, one oscillating at the frequency $\omega_{1}+\omega_{2}$ and resonant with the atomic transition at $f_{2}=0.5$ kHz, and the $\omega_{1}-\omega_{2}$ interactions on the other side of the $\omega_{1}$ component. The signatures of the higher-order interactions are visible on the right side of the resonance profile.

### 3.2. Phase Information

It was previously demonstrated that the phase of the inductive signal contains important information about the observed object/defect [23,24]. It is therefore essential that the phase information of the two-photon signal can be recovered.

This is achieved by the synchronisation of the output channels of two signal generators, SG1 and SG2. As described in Section 2, two channels (CH1 and CH2) of SG1 are synchronised with each other and with one channel (CH1) of SG2. This channel (CH1 of SG2) defines the external reference frequency $\omega_{\mathrm{ref}}$ for the lock-in amplifier that monitors the two-photon signal. An example of the output signals, *X* and *Y*, of the lock-in amplifier is shown in Figure 5 (green and orange solid lines), with the corresponding amplitude and phase (blue solid lines). These data were recorded in an unshielded environment at a Larmor frequency convenient for field stabilisation, 49.9 kHz, showing the two-photon magnetic resonance with phase information, where ϕ=arctan(Y/X). It should be noted that the frequency on the *x*-axis describes $f_{\mathrm{ref}}=f_{1}+f_{2}$. Frequency $f_{2}$ is fixed at 0.5 kHz whilst $f_{1}$ is scanned across the two-photon resonance. This is one realisation to recover phase information; it is also possible to analyse the real and imaginary parts of the FFT.

The on-resonance two-photon transition is observed at $f_{\mathrm{ref}}=49.9$ kHz, at which point $f_{2}=0.5$ kHz, $f_{1}=49.4$ kHz, and $f_{0}=f_{1}+f_{2}$ are satisfied. Another peak is also visible at 50.4 kHz. The peak occurs at a frequency when the atomic coherence is driven by an rf field where $\omega_{1}=\omega_{0}$, i.e., the single-photon condition, but this represents data that have been demodulated by the lock-in at $\omega_{\mathrm{ref}}=\omega_{1}+\omega_{2}$. The lock-in has a time constant of 10 ms and a 24 dB filter that acts as a narrow bandpass filter around $\omega_{\mathrm{ref}}$. Hence, there should be no direct single-photon component at $\omega_{1}$ demodulated at $\omega_{\mathrm{ref}}$. Consequently, there is a single-photon component within the two-photon signal. This is observed due to the finite linewidth of the magnetometer. A magnetometer with a smaller linewidth would see a reduced two-photon peak at 50.4 kHz. This component shows an rf-broadened structure of the single-photon peak due to the large amplitude of $B_{1}\left(t\right)$ [13].

### 3.3. Comparison of Single- and Two-Photon Process Efficiencies

At the fundamental level, the interaction between atoms and the rf fields through the two-photon interaction is characterised by the Rabi frequency $\Omega_{2p}=\Omega_{1}\Omega_{2}/\left(4\omega_{2}\right)$, where $\Omega_{1}=g_{F}\mu_{B}B_{1}/\hbar$ and $\Omega_{2}=g_{F}\mu_{B}B_{2}/\hbar$ are linear interaction strengths of rf fields $B_{1}$ and $B_{2}$, respectively [33,35]. This indicates that the comparison of efficiency with the linear single-photon process $\Omega_{\mathrm{sp}}$ is described by factor
(1)$$\frac{\Omega_{2p}}{\Omega_{\mathrm{sp}}}=\frac{\Omega_{1}\Omega_{2}}{4\Omega_{\mathrm{sp}}\omega_{2}}.$$

For the two-photon transition to be used experimentally, it is important for $\omega_{2}$ to be larger than the single-photon linewidth Γ. As Γ/(2π)≈50 Hz in this section, $f_{2}=500$ Hz is used. Assuming small-strength rf fields such as $\Omega_{1}=\Omega_{\mathrm{sp}}=\Omega_{2}=2\pi \left(7\;\mathrm{Hz}\right)\equiv 2$ nT where there is no rf broadening, then $\Omega_{2p}/\Omega_{\mathrm{sp}}∼0.0035$. The effective strength of the two-photon transition can be improved with an increased rf amplitude $\Omega_{2}$, or with a decreased $\omega_{2}$ as long as $\omega_{2}\gg \Gamma$. Increasing $\Omega_{2}$ is possible but can often be limited by the current source being used.

Equation (Equation 1) is verified experimentally in Figure 6. As a baseline measurement, the signal produced by a single-photon interaction was measured. This was conducted by directing the bias field along the *z*-axis and directing the coil producing $B_{1}\left(t\right)$ along the *y*-axis perpendicular to the bias field, as depicted in Figure 2b. A $\Omega_{1}=2.37$ nT 50 kHz rf field (2 $V_{\mathrm{pp}}$ directly from CH1 of SG1 to the coil along the *y*-axis in Figure 2b) produced a magnetic resonance signal with a 3.9 mV amplitude that is plotted in Figure 6. The calibration of $B_{1}$ from volts to nT was obtained by varying the amplitude $B_{1}$ and measuring the linewidth of the single-photon magnetic resonance signal. The resultant linewidths were fitted to $\Gamma =\Gamma_{0}\sqrt{1+{\left(B_{1}/B_{\mathrm{sat}}\right)}^{2}}$ [16,36], where $B_{\mathrm{sat}}=2\Gamma_{0}/\gamma_{\mathrm{cs}}$. A conversion of 1.187 V/nT at 50 kHz was obtained for this coil and a conversion of 1.096 V/nT at 50 kHz for the orthogonal coil.

For the two-photon interaction, the field producing $B_{1}\left(t\right)$ at ∼49.5 kHz had a Rabi frequency $\Omega_{1}/\left(2\pi \right)=23.7$ nT (20 $V_{\mathrm{pp}}$). Due to the inductance of the coil and ferrite, the coil producing the low-frequency rf magnetic field $B_{2}\left(t\right)$ along the bias field was also calibrated. This was conducted by reducing the bias field and observing how the signal changed with frequency. The coil producing $B_{2}\left(t\right)$ has an almost-flat frequency response from DC—50 kHz; however, it is not possible to exactly calibrate due to degradation of the single-photon atomic magnetometer performance at low frequencies. Using these numbers, a ratio of $\Omega_{2p}/\Omega_{\mathrm{sp}}=0.38$ is calculated, close to the experimentally obtained values of 0.29 in Figure 6. This analysis has shown that, despite the ability to operate at low frequencies using the two-photon transition, there is a significant drawback in terms of the reduced sensitivity of the two-photon magnetometer compared to the single-photon case, which in this section is ∼ 300 worse for the two-photon magnetometer for comparable-strength rf fields at $f_{2}=500$ Hz.

### 3.4. Inductive Measurements with Two-Photon-Based Detection

Geometrically, there are two fixed requirements to drive a two-photon coherence. It requires a linearly polarised rf field $B_{2}\left(t\right)$ directed along the static field $B_{0}$ (field along single axis) and a circularly polarised field $B_{1}\left(t\right)$ perpendicular to $B_{0}$ (field plane). The field’s polarisation is generated along the axis of a circular coil (linear polarisation equal to the sum of orthogonal circular polarisations) and does not need to be engineered, as shown in Figure 2c. Additionally, the sum (or difference) in both field frequencies must equal the operational frequency set by $B_{0}$. There is a choice between frequencies $\omega_{1}$ and $\omega_{2}$, i.e., $\omega_{1}>\omega_{2}$ or $\omega_{2}>\omega_{1}$. Here, it is beneficial to direct $B_{2}\left(t\right)$ along $B_{0}$ due to the two-photon Rabi frequency $\Omega_{2p}\propto 1/\omega_{2}$ dependence, as described in Section 3.3.

In the two-photon inductive measurements described in Ref. [32], the primary rf field $B_{2}\left(t\right)$ is directed along $B_{0}$ and the auxiliary field $B_{1}\left(t\right)$ is perpendicular to $B_{0}$. This configuration is sensitive to the primary rf field and to the secondary rf field component $B_{2}^{\prime }\left(t\right)$ parallel to the primary field. Consequently, a non-zero signal is generated over the whole area of the object.

However, information regarding an object’s composition and its defect tomography is encoded in all the components of the secondary rf field. For a flat (homogeneous) object surface, the dominant object response is typically directed along the normal surface (Figure 1). The presence of defects will result in the creation of orthogonal secondary rf field components parallel to the surface, as depicted in Figure 3a–c. Previous studies [23,24,26] have shown that, for objects whose inductive properties are dominated by electrical conductivity, it is beneficial to detect the signal generated by the components of the secondary field parallel to the surface of the tested object. In the single-photon self-compensation case (Figure 3a), no primary field is measured due to the alignment of the bias field with the primary coil, and only the secondary field is measured, leading to a high-contrast measurement.

Having $B_{0}$ perpendicular to $B_{1}\left(t\right)$ and $B_{2}\left(t\right)$ in the two-photon case (Figure 3b) provides near-equivalent functionality of the self-compensation single-photon case. The auxiliary high-frequency field $B_{1}\left(t\right)$ is far from the plate and thus does not interact with the plate. Without a defect in the plate, the configuration in Figure 3b leads to no measured signal. However, the generation of a component parallel to the plate’s surface due to the presence of a defect (Figure 3b) leads to the low-frequency-induced signal being along the bias field, leading to a high-contrast measurement of the defect. The fact that the low-frequency $B_{2}^{\prime }\left(t\right)$ field is along the bias field increases the two-photon coherence amplitude due to the $\Omega_{2p}\propto 1/\omega_{2}$ dependence. This defines the optimal geometric configuration for the inductive measurements.

The downside of using two coils for the two-photon case is securing the orthogonality of the relevant coils. The instrumentation is simplified in the single-coil two-photon case as there is only one coil that is directed perpendicular to the bias field (Figure 3c). In the single-coil case, there is now the added complication that both rf fields $B_{1}\left(t\right)$ and $B_{2}\left(t\right)$ are close to the sample during the MIT measurements. However, when inspecting the mm-deep sub-surface features as in this paper, only low-frequency rf fields (<1 kHz) can penetrate through the surface to reach the defect, as illustrated in Figure 3c.

To study inductive measurements with two-photon-based detection and compare its performance to the single-photon detection in the self-compensation arrangement, the inductive response was measured over a series of cavities (holes drilled in the side of an aluminium plate), offset from the surface of the plate at various distances [37]. The cavities are 2 mm in diameter and drilled to a length of 40 mm in the side of a 10 mm-thick plate with an area of 140 mm×140 mm. The cavities run parallel to the surface of the plate. The cavities mimic sub-surface defects or pilot holes within an object.

Figure 7 shows the amplitude and phase of the two-photon signal recorded over cavities (as depicted in Figure 3) offset by 0.5 mm (dot-dashed black line at plate position 23 mm) and 1 mm (dot-dashed red line at 63 mm) from the surface of the plate. These measurements were carried out at fixed frequencies, $f_{2}=1.5$ kHz and $f_{0}=50$ kHz, while $f_{1}$ was swept over resonance. The signatures of the cavities have a dispersive-like character due to the imperfect orthogonal positioning of the coil relative to the bias field leading to an offset, upon which the defect signals (negative on one side of the cavity and positive on the other side) add. A simple figure of merit is the signal contrast, defined by the difference between the maximum and minimum amplitude, or phase, of the feature response, e.g., at 20 mm and 25 mm for the 0.5 mm-deep cavity. The steep changes in the signal at <10 mm and >70 mm are due to the large plate edge signature.

The amplitude contrast of the observed signal reflects the depth of the feature. To confirm this observation, modelling within COMSOL 6.0 was performed using the Magnetic Field package in the frequency domain. The coil and object geometry are representative of the experiment and are described in more detail in Ref. [26]. The aluminium plate (conductivity σ=37 MS/m) contained buried cavities at four different depths (0.5 mm, 1 mm, 2 mm, and 3 mm), as in the experimental setup (Figure 3). Figure 8 shows the measured (orange circles) and modelled (blue crosses) amplitudes of the inductive measurement signals as a function of cavity depth. The COMSOL data are analysed as explained in Ref. [26] for an atomic magnetometer in the self-compensation configuration. Close agreement between the experimental and modelled data can be observed. Deviations between the measured and simulated data could exist due to the plate being tilted and thus leading to uncertainty regarding the distance from the ferrite core to the recess.

In the previous subsection, a comparison was presented regarding the strengths of the single- and two-photon interactions between the atoms and the rf field. This comparison reflects the differences in the detection efficiencies through these two processes. As pointed out in the Introduction, at low operating frequencies, the stabilisation of the bias magnetic field becomes challenging. Possible instabilities would be reflected in a broadened magnetic resonance and effectively lower signal amplitude. This is shown via comparisons of the inductive measurements performed using the two techniques. For a fair comparison, these measurements were performed with the same $f_{\mathrm{sp}}=f_{2}=2$ kHz. For single-photon-based detection, this requires setting the bias field strength to 570 nT, below which the unshielded sensor is significantly affected by environmental magnetic field noise.

Figure 9 shows the signal amplitudes recorded over previously described cavities with the sensor using the two-coil two-photon (orange line) and single-photon interaction (self-compensation mode with one coil, dotted-blue line). The contrasts of the cavities’ signatures are noticeably smaller in the case of the single-photon-based detection, which could be caused by bias field instabilities (deviation from self-compensation geometry). It needs to be pointed out that the primary rf field in both cases has the same frequency, i.e., penetration depth, and the differences in amplitude reflect issues related to the overall operating frequency, i.e., stability of the bias magnetic field. The data were also recorded for the two-photon process generated by a single coil, demonstrating a comparable response. For all these measurements, the coils are equidistant from the sensor to maintain their respective field amplitudes. The datasets are normalised relative to the average of the central 20 data points around position 41.7 mm between the two recesses.

Deciding on when to use the single-photon or two-photon magnetometer depends on the depth of the recess under investigation. For example, operating in the single-photon self-compensation configuration at 10 kHz would be possible in a harsh, unshielded environment, but such rf fields have a small skin depth of 0.8 mm in aluminium. This means that detecting >1 mm-deep recesses becomes very difficult and is swamped by background signals as large primary fields are required, making the magnetometer more sensitive to misalignments of the coil and to surface effects, such as the signal generated by the edge of the plate. Operating at 500 Hz in the two-photon configuration, however, is practically straightforward and allows one to convert more of the primary field into a secondary field as the skin depth at this frequency is 3.7 mm, avoiding the possibility of the primary field washing out the recess signal. Fundamentally, sub-surface defects should be investigated with the lowest possible frequency attainable where the sensor can operate with high sensitivity.

## 4. Conclusions

The present work demonstrated the two-photon method as a practical realisation of low-frequency inductive measurements. Compared with the single-photon method, the signal strength is a factor of 300 smaller at $f_{2}=500$ Hz. However, the data in Figure 9 show a relative improvement in the measurement contrasts at 2 kHz. This is close to the practical limit for single-photon operation due to the limitations in ambient field stabilisation. In the experimental setup presented here, the advantage of the two-photon method comes when operating below this limit, which is required to detect the 2 mm and 3 mm deep cavities shown in Figure 8. The signatures of the deeper recesses are unclear using the single-photon method at 2 kHz. The low contrast signal, blurred by instabilities, is due to operation at low $B_{0}$. Additionally, when the system is not limited by excitation current, it is possible to mitigate the reduced efficiency of the two-photon method by increasing the strengths $B_{1}$ and $B_{2}$.

In summary, this paper discussed the practicalities and limitations of operating the rf atomic magnetometer in the two-photon configuration for inductive measurements and presented several novel experimental realisations. (1) Systematic observations of the magneto-optical signals in a magnetically shielded environment enabled the observation of the higher-order spectral components produced by interactions between the atoms and two rf fields with distinct frequencies. (2) It is shown that the phase information of the rf fields is recoverable through lock-in signal detection. This is relevant for the implementation of two-photon detection in inductive measurements, where the tomographic information about possible defects within the object is encoded in both the amplitude and phase of the rf field. (3) To enhance the signal contrast of the inductive measurements with the two-photon measurement, a self-compensation configuration is demonstrated whereby a signal is only measured in the presence of a defect. (4) This measurement configuration is also realised with only a single rf coil, reducing the experimental complexity. (5) Despite the downsides of the reduced sensitivity of the two-photon rf magnetometer versus the single-photon rf magnetometer, this paper demonstrates the critical role that two-photon magnetometers can play in non-destructive testing in the future.

## Figures and Tables

**Figure 1 sensors-24-06657-f001:**
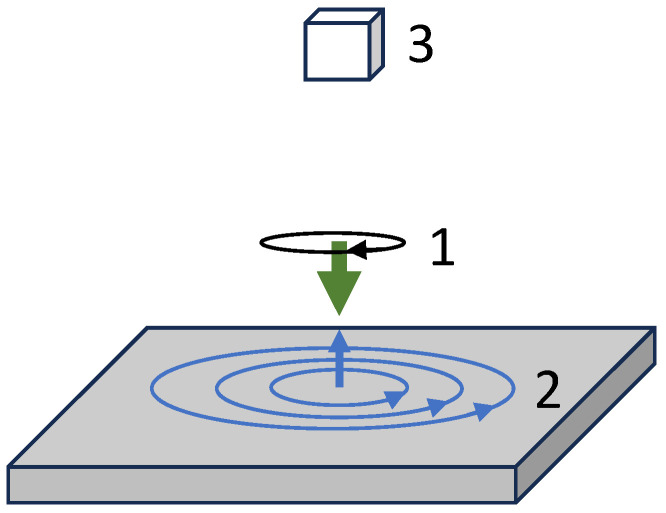
Model of the generic configuration of an inductive measurement. An excitation field (1), the so-called primary rf field and represented by the green arrow, drives the object response (2), which in this case is the generation of eddy currents denoted by the blue circles. These produce a secondary rf field, represented by a blue arrow. The resultant field is detected by a sensor (3), depicted by the white box.

**Figure 2 sensors-24-06657-f002:**
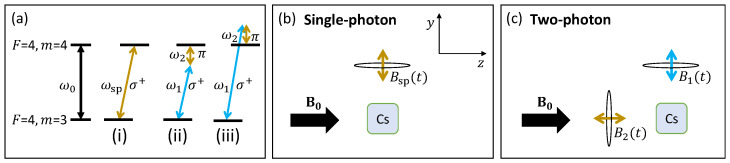
(**a**) Detection of the rf field with an atomic magnetometer is performed by monitoring the amplitude and phase of the atomic coherence driven by the rf field between Zeeman sublevels of the F=4 caesium ground state. For simplicity, only two sublevels are shown. In the single-photon case in [a(i)], the transition frequency $\omega_{0}$ is tuned into resonance with the detected rf field frequency $\omega_{\mathrm{sp}}$ by adjusting the bias magnetic field $B_{0}$. The sensor detects only the circularly polarised component ($\sigma^{+}$) of the rf field. In the two-photon case, the atomic coherence is driven by two rf fields. The resonance condition is met by the [a(ii)] sum or [a(iii)] difference in the field frequencies. Selection rules set the conditions for the polarisations of the fields. The gold coloured arrows represent the field that drives the response of interest (low frequency) from the object relevant to the inductive measurements. The choice of frequency and field polarisation used is described in Section 3.4. (**b**) The single-photon resonance condition can be satisfied when an rf field $B_{1}\left(t\right)$ is applied perpendicularly to $B_{0}$ with $\omega_{\mathrm{sp}}=\omega_{0}$. (**c**) In the two-photon configuration, an extra rf field $B_{2}\left(t\right)$ is required along the bias field axis such that a two-photon transition can be achieved.

**Figure 3 sensors-24-06657-f003:**
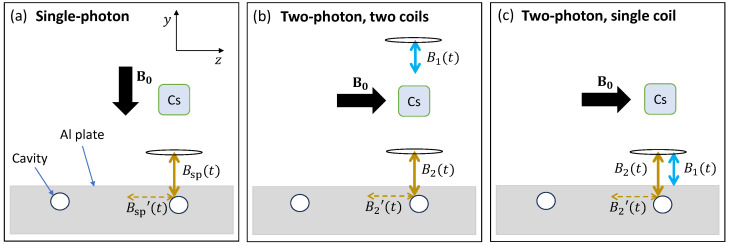
MIT experimental setups for the (**a**) single-photon, (**b**) two-photon two-coil, and (**c**) two-photon single-coil configurations. The gold coloured arrows represent the field that drives the response of interest (low frequency) from the object relevant to the inductive measurements. (**a**) In the single-photon self-compensation case, the bias field is directed along the primary field $B_{\mathrm{sp}}\left(t\right)$ (double-ended gold arrow), and the magnetometer is sensitive to secondary fields $B_{\mathrm{sp}}^{\prime }\left(t\right)$ in the 2D plane perpendicular to $B_{0}$. (**b**) In the two-photon two-coil configuration, the high-frequency auxiliary coil producing $B_{1}\left(t\right)$ is far from the plate. The low-frequency rf field $B_{2}\left(t\right)$ can penetrate through the material due to its large skin depth, and the secondary field $B_{2}^{\prime }\left(t\right)$ induced parallel to the surface of the plate is measured by the sensor. The optimal geometric configuration is chosen due to the $1/\omega_{2}$ amplitude dependence of the two-photon coherence, described in Section 3.3. (**c**) In the two-photon single-coil case, both frequency components come from the same coil. Only the low-frequency component will produce a secondary field $B_{2}^{\prime }\left(t\right)$ along the bias field due to the attenuation of the high-frequency rf field at the object’s surface.

**Figure 4 sensors-24-06657-f004:**
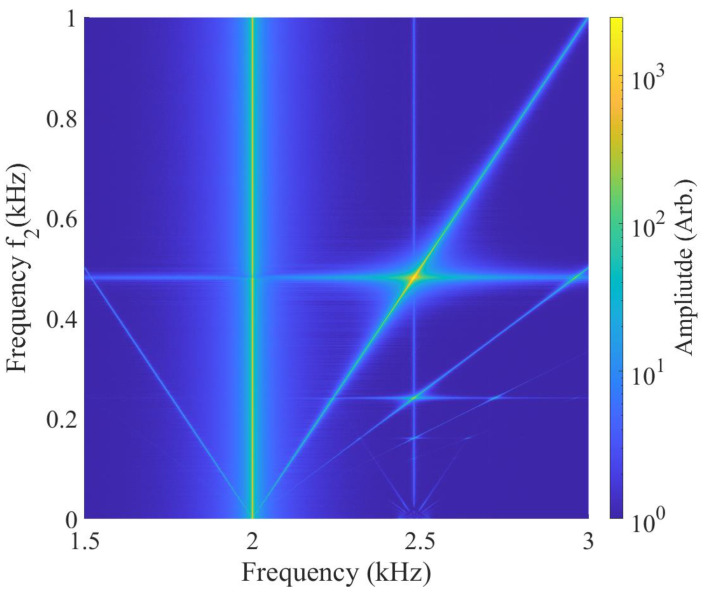
FFTs of the polarisation rotation signal recorded as $f_{2}$ are scanned over the 0–1 kHz frequency range, whilst $f_{1}$ remains fixed at 2 kHz in shielded conditions. The two-photon profiles are represented by the diagonal lines, $f_{1}\pm f_{2}$. Atomic shot noise produces a weak signal at the resonant frequency $f_{0}=2.48$ kHz. The two-photon resonance can be observed when $f_{0}=f_{1}+f_{2}$.

**Figure 5 sensors-24-06657-f005:**
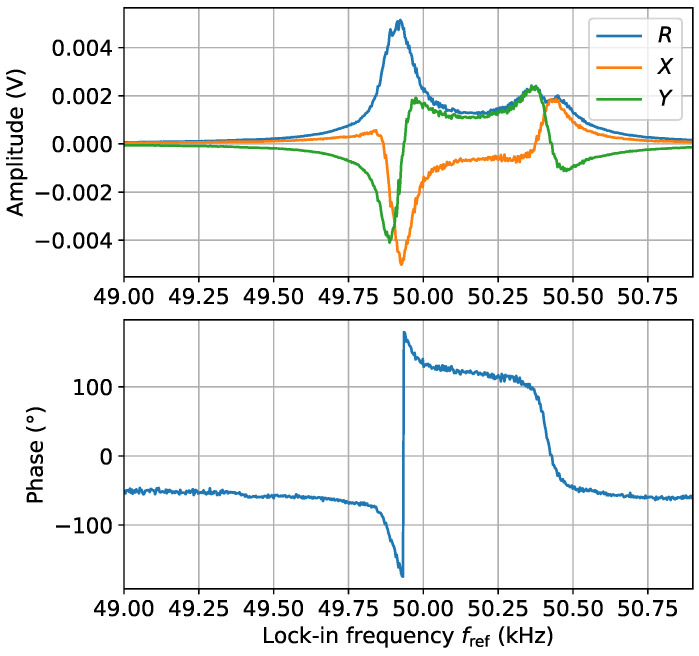
The in-phase (*X*), quadrature (*Y*), and magnitude (*R*) components of the lock-in are monitored during the two-photon resonance signal, demonstrating that phase information ϕ=arctan(Y/X) can be obtained in a two-photon measurement. The two-photon transition can be observed at $f_{\mathrm{ref}}=f_{0}=49.9$ kHz ($f_{2}=0.5$ kHz and $f_{1}=49.4$ kHz). At $f_{\mathrm{ref}}=50.4$ kHz ($f_{2}=0.5$ kHz and $f_{1}=49.9$ kHz), the single-photon transition is driven by $f_{1}$, which is then contained within the two-photon signal at 50.4 kHz. The “double peaks” in the 50.4 kHz data are due to the high rf broadening, which occurs when $B_{1}$ is large.

**Figure 6 sensors-24-06657-f006:**
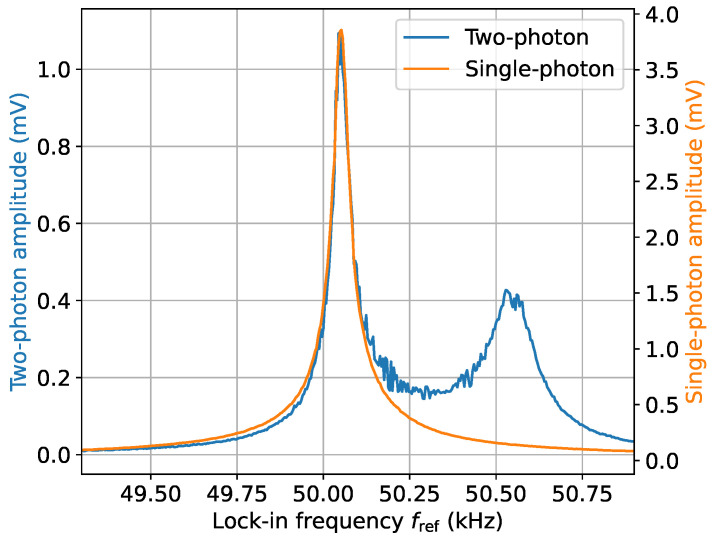
Determining the efficiency of the two-photon transition versus the single-photon transition. The single-photon measurement as in Figure 2b used the settings $f_{2}=0$ kHz, $f_{\mathrm{ref}}=f_{\mathrm{sp}}$, and $B_{\mathrm{sp}}=2.37$ nT, and the two-photon measurement as in Figure 2c used $f_{2}=$ 0.5 kHz, $f_{\mathrm{ref}}=f_{1}+f_{2}$, $B_{1}=23.7$ nT, and $B_{2}=21.9$ nT. This enables a comparison of the single-photon and two-photon efficiencies to be undertaken.

**Figure 7 sensors-24-06657-f007:**
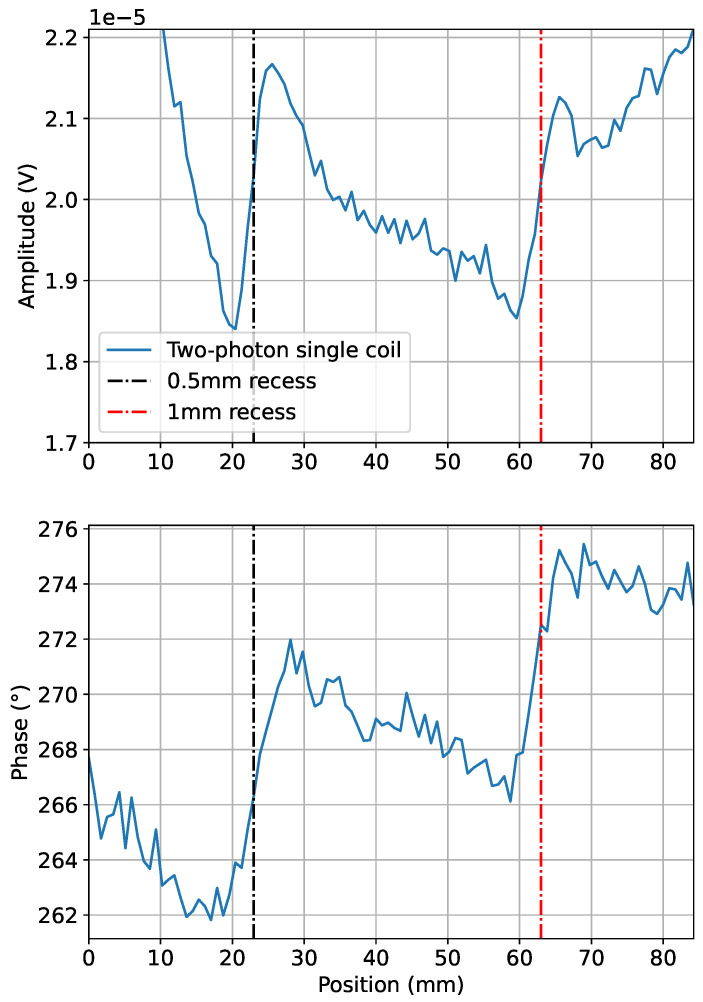
Two-photon ($f_{2}=1.5$ kHz, $f_{1}=48.5$ kHz, and $f_{0}=50$ kHz) single-coil linescan data over the 0.5 mm and 1 mm cavities in the Al pilot hole plate. The amplitude and phase are plotted as the plate is moved under the excitation coil. This demonstrates the capability of obtaining phase information from a two-photon measurement during MIT measurements.

**Figure 8 sensors-24-06657-f008:**
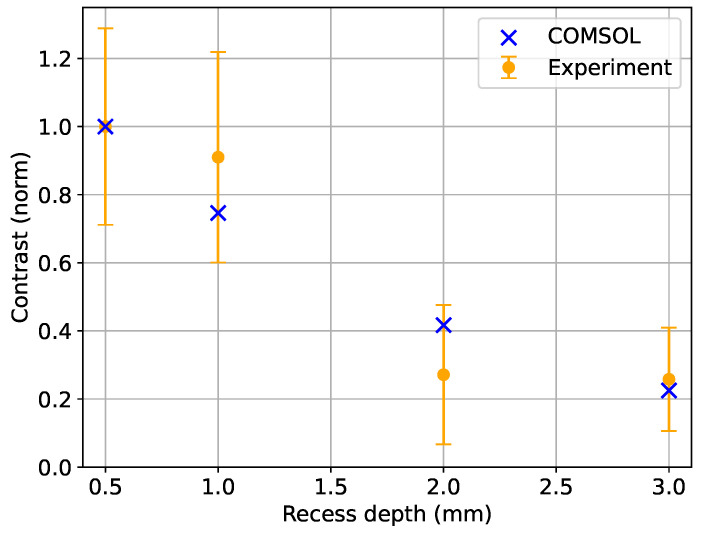
The experimentally obtained amplitudes (blue dots) were obtained for four different cavity depths (0.5 mm, 1 mm, 2 mm, and 3 mm) in the two-photon single-coil configuration with $f_{2}=500$ Hz and $f_{0}=50$ kHz. The COMSOL data (orange crosses, $f_{2}=500$ Hz) were performed using the same setup as in Ref. [26] but for the sub-surface cavities described in this paper instead of the open recess used in Ref. [26]. The contrast was normalised by the signal from the shallowest cavity for both the experimental and modelled datasets. Each contrast data point was calculated as the difference between the maximum and minimum amplitudes in a linescan, repeated to determine means and standard deviations. Each error bar is the propagation of the standard error of the mean (SEM) through the contrast and normalisation calculations.

**Figure 9 sensors-24-06657-f009:**
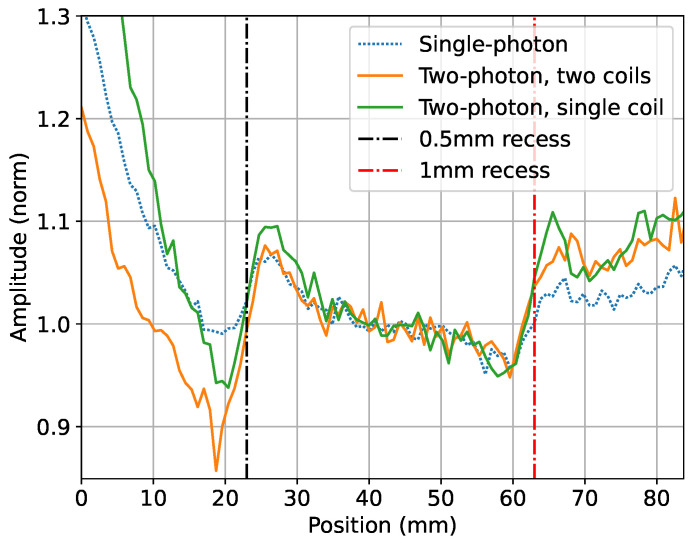
Comparison of MIT measurements over 0.5 mm and 1 mm deep cavities for the single-photon ($f_{0}=2$ kHz and $f_{\mathrm{sp}}=2$ kHz in Figure 3a), the two-photon single-coil ($f_{0}=50$ kHz, $f_{2}=2$ kHz, and $f_{1}=48$ kHz in Figure 3b), and the two-photon two-coil ($f_{0}=50$ kHz, $f_{2}=2$ kHz, and $f_{1}=48$ kHz in Figure 3c) configurations.

## Data Availability

The data that support the findings of this study are available from the corresponding author upon reasonable request.

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
