# Peer review of "Performance of a Radio-Frequency Two-Photon Atomic Magnetometer in Different Magnetic Induction Measurement Geometries"

_sensors, 2024, doi:10.3390/s24206657_

Round 1

Reviewer 1 Report

Comments and Suggestions for Authors

The authors present a systematic study of the radio-frequency two-photon atomic magnetometer in both magnetically shielded and unshielded environments. The approach offers advantages for ultra-low frequency inductive measurements, particularly in an unshielded setup. The magnitude and phase information of the measured signal are analyzed comprehensively. They also compare the efficiency of single- and two-photon processes and different measurement configurations theoretically and experimentally. The manuscript is well-written, with a clear presentation, and the study is thorough. The results appear suitable for publication in the Sensors journal. Nonetheless, I recommend addressing two specific aspects:

1. In two-photon process, the rf coils may have different configurations.  It would be beneficial to provide the guidelines for designing the direction, phase and amplitude of the rf fields.

2. Compared to the single-photon case, the two-photon magnetometer shows reduced sensitivity. It could be more informative if the authors provide a rough parameters range or specific parameter settings where the two-photon atomic magnetometer yields comparable or superior results in low-frequency inductive measurements.

Author Response

  1. In two-photon process, the rf coils may have different configurations.  It would be beneficial to provide the guidelines for designing the direction, phase and amplitude of the rf fields.

We have added the paragraph below at the start of the section on inductive measurements to help explain the geometrical considerations in the experimental setup. One point of confusion might be that the polarisation of the rf fields do not need to be engineered. These polarisations are sourced freely by using circular coils driven by a sinusoidal current. Such a coil generates a linearly polarised rf field along the coil axis, which is composed of two orthogonal circular fields.

‘Geometrically there are two fixed requirements to drive a two-photon coherence. It requires a linearly polarised rf field $\textbf{B}_{2}(t)$ directed along the static field $\textbf{B}_{0}$ (field along single axis) and a circularly polarised field $\textbf{B}_{1}(t)$ perpendicular to $\textbf{B}_{0}$ (field plane). The field's polarisation is generated along the axis of a circular coil (linear polarisation equal to the sum of orthogonal circular polarisations) and does not need to be engineered, as shown in Fig.~\ref{fig:energyleveldiagram}c. Additionally, the sum (or difference) of both field's frequencies must equal the operational frequency set by $B_{0}$. There is choice over the frequencies $\omega_1$ and $\omega_2$, i.e., $\omega_1>\omega_2$  or $\omega_2>\omega_1$. Here it is beneficial to direct $\textbf{B}_2(t)$ along $\textbf{B}_0$ due to the two-photon Rabi frequency $\Omega_{\text{2p}}\propto 1/\omega_2$ dependence, as described in Sec.~\ref{sec:efficiencies}.’

  1. Compared to the single-photon case, the two-photon magnetometer shows reduced sensitivity. It could be more informative if the authors provide a rough parameters range or specific parameter settings where the two-photon atomic magnetometer yields comparable or superior results in low-frequency inductive measurements.

We thank for the reviewer for highlighting this omission. This is partially answered in the data we present in Fig. 9, which shows the increase in contrast of the two-photon signal representing the defect relative to the single photon case. The challenge with direct comparison is the benefits of using the two-photon method is related to operational conditions, i.e., performance in the presence of magnetic field noise.

The following discussion has been added to the end of section 3.4 on inductive measurements.

‘Deciding on when to use the single-photon or two-photon magnetometer depends on the depth of the recess under investigation. For example, operating in the single-photon self-compensation configuration at 10~kHz would be possible in a harsh, unshielded environment, but such rf fields have a small skin depth of 0.8~mm in aluminium. This means that detecting $>1$~mm-deep recesses becomes very difficult and is swamped by background signals, as large primary fields are required, making the magnetometer more sensitive to misalignments of the coil and to surface effects, such as the signal generated by the edge of the plate. Operating at 500~Hz in the two-photon configuration, however, is practically straightforward and allows one to convert more of the primary field into a secondary field, as the skin depth at this frequency is 3.7~mm, avoiding the possibility of the primary field washing out the recess signal. Fundamentally, sub-surface defects should be investigated with the lowest possible frequency attainable where the sensor can operate with high sensitivity.’

We have added the text below to the conclusions, as well as providing a more general discussion of the manuscript as suggested by the second reviewer.

‘The work presented demonstrates the two-photon method as a practical realisation of low-frequency inductive measurements. Compared with the single-photon method the signal strength is a factor of 300 smaller at $f_2=500$~Hz. However, the data in Fig.~\ref{fig:MIT_TwoPhotonSingle_2PTwo_SingleP} shows a relative improvement in the measurement contrasts at 2~kHz. This is close to the practical limit for single-photon operation due to the limitations in ambient field stabilisation. In the experimental setup presented here, the advantage of the two-photon method comes when operating below this limit as is required to detect the 2~mm and 3~mm deep cavities shown in Fig.~\ref{fig:CoilGeometries}. Signatures of the deeper recesses are unclear using the single-photon method at 2~kHz. The low contrast signal, blurred by instabilities, is due to operation at low $\textbf{B}_0$. Additionally, when the system is not limited by excitation current it is possible to mitigate the reduced efficiency of the two-photon method by increasing the strengths $B_1$ and $B_2$.’

Reviewer 2 Report

Comments and Suggestions for Authors

sensors-3241185

Review of an article:

“Performance of a radio-frequency two-photon atomic magnetometer in different magnetic induction measurement geometries”

by Lucas M Rushton, Laura M Ellis, Jake D Zipfel, P. Bevington, Witold Chalupczak

Round 1

The authors showed that: (a) Two-photon interaction in rf atomic magnetometers improves inductive measurements, (b) The method also demonstrates the retrieval of two-photon phase information.

I think minor revisions are needed.

The following is a list of specific concerns.

1.     Even though, authors try to show novelty as an alphabetical list. The novelty of this article is still unclear.

2.     I highly recommend giving a separate paragraph with an aim of this article.

3.     The introduction section is too large, I think all information should rearrange. Please refine the source information.

4.     Please use a standard IMRAD type of all articles. See recommendation of the journal.

  • Discussion: Authors should discuss the results and how they can be interpreted in perspective of previous studies and of the working hypotheses. The findings and their implications should be discussed in the broadest context possible and limitations of the work highlighted. Future research directions may also be mentioned. This section may be combined with Results.

5.     Please provide possible configurations of the radio-frequency atomic magnetometer with measurement geometries.

6.     For figure 8 provide error bar for data comparison.

7.     Please, provide the proof of reproducibility of the results.

Author Response

  1. Even though, authors try to show novelty as an alphabetical list. The novelty of this article is still unclear.

To highlight the novelties, they have been listed alphabetically as noted by the reviewer. These points are also introduced the abstract. For clarity we have added a comment confirming that these have previously not been studied or presented in other works.

  1. I highly recommend giving a separate paragraph with an aim of this article.

We have expanded on the motivation in the section where we outline the novelties of this manuscript.

‘This work highlights the strengths and weaknesses of the two-photon method in the applications relevant to inductive measurements. There are geometrical requirements in generating the two-photon coherence and preferential choice of frequencies of the two fields. First, there is an outline of the experimental setup, detailing the recovery of phase information. Then, the measured signal and its components are described, before discussing the efficiency of the two-photon signal relative to the single-photon measurement. The final section details its operation when applied to inductive measurements and explains the relevant configurations to optimise the method's performance. A general discussion summarising the paper's findings is given in the conclusions.’ 

  1. The introduction section is too large, I think all information should rearrange. Please refine the source information.

The topic discussed in this manuscript covers two relatively distant areas: (quantum) atomic physics and (classical) inductive measurements. An introduction section was planned as an explanation of all relevant concepts (sometimes basic within particular field), in order to make presented arguments understandable for readers representing researchers from these two areas.

The introduction has been structured to motivate the relevance of the rf magnetometer in NDT measurements and the need for low frequency operation of the rf magnetometer. Additionally, the two-photon method needs to be clearly introduced and described with respect to the standard single photon case. Hence, we believe the introduction is a suitable length to adequately describe the topics relevant to this manuscript.

  1. Please use a standard IMRAD type of all articles. See recommendation of the journal.

We believe this model has been followed for this manuscript. The manuscript introduces the relevant concepts in a structured manner; the measured signal and its components; the efficiency of driving the two-photon coherence; how phase information can be retrieved; and its application to NDT with magnetic induction. The discussion of these themes is included in the relevant sections. The general discussion of the manuscript provided in the conclusions has been expanded with the text below.

The following discussion has been added to the end of section 3.4 on inductive measurements.

‘Deciding on when to use the single-photon or two-photon magnetometer depends on the depth of the recess under investigation. For example, operating in the single-photon self-compensation configuration at 10~kHz would be possible in a harsh, unshielded environment, but such rf fields have a small skin depth of 0.8~mm in aluminium. This means that detecting $>1$~mm-deep recesses becomes very difficult and is swamped by background signals, as large primary fields are required, making the magnetometer more sensitive to misalignments of the coil and to surface effects, such as the signal generated by the edge of the plate. Operating at 500~Hz in the two-photon configuration, however, is practically straightforward and allows one to convert more of the primary field into a secondary field, as the skin depth at this frequency is 3.7~mm, avoiding the possibility of the primary field washing out the recess signal. Fundamentally, sub-surface defects should be investigated with the lowest possible frequency attainable where the sensor can operate with high sensitivity.’

We have added the text below to the conclusions, as well as providing a more general discussion of the manuscript as suggested by the second reviewer.

‘The work presented demonstrates the two-photon method as a practical realisation of low-frequency inductive measurements. Compared with the single-photon method the signal strength is a factor of 300 smaller at $f_2=500$~Hz. However, the data in Fig.~\ref{fig:MIT_TwoPhotonSingle_2PTwo_SingleP} shows a relative improvement in the measurement contrasts at 2~kHz. This is close to the practical limit for single-photon operation due to the limitations in ambient field stabilisation. In the experimental setup presented here, the advantage of the two-photon method comes when operating below this limit as is required to detect the 2~mm and 3~mm deep cavities shown in Fig.~\ref{fig:CoilGeometries}. Signatures of the deeper recesses are unclear using the single-photon method at 2~kHz. The low contrast signal, blurred by instabilities, is due to operation at low $\textbf{B}_0$. Additionally, when the system is not limited by excitation current it is possible to mitigate the reduced efficiency of the two-photon method by increasing the strengths $B_1$ and $B_2$.’

  1. Please provide possible configurations of the radio-frequency atomic magnetometer with measurement geometries.

We appreciate that further clarity can be provided here and we have addressed this comment in our response to the first reviewer, copied below.

‘Geometrically there are two fixed requirements to drive a two-photon coherence. It requires a linearly polarised rf field $\textbf{B}_{2}(t)$ directed along the static field $\textbf{B}_{0}$ (field along single axis) and a circularly polarised field $\textbf{B}_{1}(t)$ perpendicular to $\textbf{B}_{0}$ (field plane). The field's polarisation is generated along the axis of a circular coil (linear polarisation equal to the sum of orthogonal circular polarisations) and does not need to be engineered, as shown in Fig.~\ref{fig:energyleveldiagram}c. Additionally, the sum (or difference) of both field's frequencies must equal the operational frequency set by $B_{0}$. There is choice over the frequencies $\omega_1$ and $\omega_2$, i.e., $\omega_1>\omega_2$  or $\omega_2>\omega_1$. Here it is beneficial to direct $\textbf{B}_2(t)$ along $\textbf{B}_0$ due to the two-photon Rabi frequency $\Omega_{\text{2p}}\propto 1/\omega_2$ dependence, as described in Sec.~\ref{sec:efficiencies}.’

  1. For figure 8 provide error bar for data comparison.

We thank the reviewer for this comment, and have addressed this now. Each contrast data point was calculated as the difference of the maximum and minimum amplitudes in a linescan. Each linescan was repeated multiple times, from which averages and standard deviations for each pixel could be determined. The standard deviations were divided by the sqrt(number of averages) to get a standard error of the mean (SEM), and these values were propagated through the contrast calculation (errors added in quadrature) and the normalisation (fractional errors added in quadratue) to get the final error bars. The following was added to the figure caption of Fig. 8:

‘Each contrast data point was calculated as the difference of the maximum and minimum amplitudes in a linescan, repeated to determine means and standard deviations. Each error bar is the propagation of the standard error of the mean (SEM) through the contrast and normalisation calculations.’

  1. Please, provide the proof of reproducibility of the results.

We regularly repeat measurements and record similar data over long periods of time, often months between successive measurements. These show perfect agreement through systematic studies, like those presented in this manuscript.

This is an unusual request but we have provided the data in the attachment (Date240116_RepeatMeasurements.docx) to show the reproducibility of the data recorded in Fig. 7, which shows repeat spatial linescans over the aluminium object with subsurface cavities.

We present data recorded in the single-photon configuration, which is the standard method for operating our measurement. The data recorded in this way agrees with previous publications, e.g. refs [23,24,28,29,37], and we have shown that data recorded in the two-photon method agrees with the single-photon data.
